# Effect of a Laparoscopic Donor Nephrectomy in Healthy Living Kidney Donors on the Acute Phase Response Using Either Propofol or Sevoflurane Anesthesia

**DOI:** 10.3390/ijms26115196

**Published:** 2025-05-28

**Authors:** Baukje Brattinga, Honglei Huang, Sergei Maslau, Adam M. Thorne, James Hunter, Simon Knight, Michel M. R. F. Struys, Henri G. D. Leuvenink, Geertruida H. de Bock, Rutger J. Ploeg, Benedikt M. Kessler, Gertrude J. Nieuwenhuijs-Moeke

**Affiliations:** 1Department of Surgery, University Medical Center Groningen, University of Groningen, 9713 GZ Groningen, The Netherlands; 2Nuffield Department of Surgical Sciences, Oxford Biomedical Research Centre, University of Oxford, Oxford OX3 7FZ, UK; 3Department of Anesthesiology, University Medical Center Groningen, University of Groningen, 9713 GZ Groningen, The Netherlands; 4Department of Basic and Applied Medical Sciences, Ghent University, 9000 Gent, Belgium; 5Department of Epidemiology, University Medical Center Groningen, University of Groningen, 9713 GZ Groningen, The Netherlands; 6Target Discovery Institute, Centre for Medicines Discovery, Nuffield Department of Medicine, University of Oxford, Oxford OX1 2JD, UK

**Keywords:** surgical stress response, propofol, sevoflurane, anesthesia, living kidney donors, laparoscopic surgery

## Abstract

Surgical trauma elicits a complex inflammatory stress response, contributing to postoperative morbidity and recovery variability. This response is influenced by patient-specific factors and surgical and anesthetic techniques. To isolate the impact of anesthesia on the acute phase response, we investigated plasma proteomic changes in a uniquely homogeneous cohort of healthy, living kidney donors (*n* = 36; propofol = 19; sevoflurane = 17) undergoing laparoscopic donor nephrectomy. Proteomic profiling of plasma samples collected preoperatively and at 2 and 24 h postoperatively revealed 633 quantifiable proteins, of which 22 showed significant perioperative expression changes. Eight proteins exhibited over two-fold increases, primarily related to the acute phase response (CRP, SAA1, SAA2, LBP), tissue repair (FGL1, A2GL), and anti-inflammatory regulation (AACT). These changes were largely independent of anesthetic type, though SAA2 and MAN1A1 showed anesthetic-specific expression. The upregulation of these proteins implicates the activation of immune pathways involved in host defense, tissue remodeling, and inflammation resolution. Our findings provide a molecular reference for the surgical stress response in healthy individuals and highlight candidate biomarkers for predicting and managing postoperative outcomes. Understanding these pathways may support the development of strategies to mitigate surgical stress and enhance recovery, particularly in vulnerable patient populations.

## 1. Introduction

Each year, over 4.2 million patients die within 30 days after a surgical procedure, making postoperative mortality, with 7.7% occurrence, the third leading cause of death after ischemic heart disease and stroke [1]. In addition, about 20% of patients develop postoperative complications, which are linked to poorer long-term outcomes, longer hospital admissions, increased costs and a burden to society [2]. Even after complete recovery, 20–40% of patients experience a decline in functional capacity or quality of life [3]. Consequently, the global disease burden of surgical procedures is substantial.

Poor outcomes after surgery are most likely the result of a complex interplay between the underlying disease of the patient, his or her comorbidities, physiological reserve capacity, genetic predisposition, and the individual biological stress response to tissue injury. Surgery induces a stress response due to both direct (e.g., tissue dissection, traction, and heating) and indirect (e.g., hypoperfusion, acidosis, ad ischemia) tissue injury, leading to hormonal, metabolic, and immunological changes [4,5,6]. Prolonged elevation of cytokines (e.g., interleukin(IL)-1b, IL-6, IL-8 and tumor necrosis factor alpha (TNF-α)), hormones (e.g., cortisol), acute phase proteins (C-reactive protein (CRP)), and activation of the coagulation system after surgery are likely associated with an increased risk of postoperative complications [7,8,9]. Increasing evidence shows that intraoperative interventions like maintenance of normothermia, prevention of acidosis, optimization of fluid therapy, using minimally invasive surgical techniques, and the choice of anesthetics can modulate the stress response and reduce the incidence of postoperative complications [10,11,12,13,14,15]. Furthermore, it is known that anesthetic agents have immunomodulatory, anti-necrotic, and anti-apoptotic effects [16,17]. Both the intravenous anesthetic propofol and the inhalational anesthetic sevoflurane have protective and anti-inflammatory effects and are routinely used. Reduced levels of IL-1, IL-6, and IL-8, as well as significant reduction in oxidative stress have been found in propofol anesthesia [18,19,20,21], whilst inhibition of the production of IL-1b, IL-6, and apoptosis regulatory proteins such as cytochrome C and caspase 3 have been reported for sevoflurane [22].

Unraveling the surgical stress response and identifying pre- and intraoperative factors that interfere with this response is important as it may reveal modifiable factors that could improve surgical outcomes. Unfortunately, most patients undergoing a substantial surgical procedure have an indication due to an acute or chronic disease and often suffer from comorbidities that will interfere with perioperative stress responses and affect outcomes. This makes it difficult to evaluate the bespoke effect of the proper surgical procedure on stress at a molecular level. The living donor procedure in kidney transplantation is a unique ‘model’ as this cohort concerns healthy subjects (and not patients), without any or minimal comorbidity, who are exposed to an intermediate-risk surgical procedure (laparoscopic donor nephrectomy). In this study, we assessed the stress response caused by the surgical procedure by analyzing proteomic profiles from sequential plasma samples in the perioperative period. We also compared the effects of using either propofol- or sevoflurane-based anesthesia. Gaining a better insight into the effect of surgery and anesthesia in healthy individuals might enhance our understanding of the surgical stress response and create reference data for future comparison in other patient categories.

## 2. Results

### 2.1. Patient Characteristics

Table 1 summarizes donor baseline characteristics and clinically relevant perioperative parameters. Donors had a low comorbidity score and were considered fit to donate a kidney by a clinical team. The Charlson Comorbidity Index was comparable between the propofol (PROP) group and the sevoflurane (SEVO) group (1 (0–2) vs. 1 (0–1), *p* = 0.62). The most common comorbidities were hypertension and hypercholesterolemia, combined as cardiovascular comorbidity in Table 1. There was a higher incidence of cardiovascular comorbidity in the SEVO group compared to the PROP group (5 (29%) vs. 1 (5%), *p* = 0.080). Medications used preoperatively in both groups were beta-blockers (SEVO 2), angiotensin (AT)-II antagonists (SEVO 2), statins (PROP 1, SEVO 3), proton pump inhibitors (PROP 3), platelet aggregation inhibitors (SEVO 2), and antidepressants (PROP 1). The duration of the surgical procedure was comparable between groups. Donors anesthetized with sevoflurane showed a higher average bispectral index (BIS) value during the procedure (45 (6) vs. 39 (7), *p* = 0.012). Donors anesthetized with sevoflurane more frequently received a bolus of ephedrine after induction of anesthesia compared to patients anesthetized with propofol (17 (100%) vs. 13 (68%), *p* = 0.020). This occurred predominantly after the induction of anesthesia. No extended hypotensive periods were observed and none of the donors received vasoactive medication on a continuous basis. Intraoperative hemodynamic profiles over time of mean arterial pressure and heart rate were comparable between groups (data available in the data repository DataverseNL). In all donors, remifentanil was started at 2 ng/mL effect site concentration (Cet) and the average Cet of remifentanil during the procedure was significantly higher in the PROP than SEVO group (3.2 (0.86) vs. 2.6 (0.63); *p* = 0.02). The amount of fluid administered intraoperatively was similar, and Ringer’s lactate was used. No colloids were used. Arterial blood sampling at the time of retrieval of the kidney showed comparable pH, oxygen tension, and levels of hemoglobin. Lactate levels at that time point were higher in the SEVO than in the PROP group (1.7 (0.7) vs. 1.3 (0.5); *p* = 0.04). One patient in the PROP group developed pneumonia in the postoperative period, which was treated with antibiotics. One patient in the PROP group developed hypokalemia, which resolved spontaneously. The length of hospital stay was comparable between groups: 5–7 days for the PROP group and 5–7.5 days for the SEVO group (*p* = 0.91).

### 2.2. Proteomics Profiles Dependent of Anesthetic Regimen

Quantitative proteomics analysis resulted in the detection of 633 plasma proteins. A subset of 22 proteins showed statistically significant changes in expression levels between time points T2 (24 h after surgery) vs. T0 (baseline) (Multiple Testing Correction (MTC) *p* < 0.05, Table 2). When comparing expression changes in these 22 proteins between groups, PROP anesthesia induced a change in the abundance of 15 proteins (Figure 1): SEVO in 18 proteins (Figure 2). Of the 15 proteins that reached statistical significance in PROP (MTC *p* < 0.05), 7 showed over a 2-fold increase. Similarly, out of 18 significantly increased proteins in SEVO (MTC *p* < 0.05), 7 showed over a 2-fold increase. Changes in plasma protein levels that were independent of the anesthesia type included 11 proteins that were consistent in the direction and magnitude of change (Table 2). These changes were statistically significant after MTC and at a threshold of at least *p* < 0.05, and often lower, reflecting increased stringency. Typically, the levels of proteins were increased following surgical intervention, with the exception of plasma serine protease inhibitor (IPSP), which showed a >1.5-fold decrease (*p*-value < 0.01 MTC). There were 6 proteins with greater than 2-fold changes, independent of anesthesia type: CRP (>35-fold); fibrinogen-like protein-1 (FGL-1, >10 fold); serum amyloid A-1 protein (SAA1, >9-fold); LPS binding protein (LBP, >4-fold); leucine-rich alpha-2-glycoprotein (A2GL, >2-fold); alpha-1-antichymotrypsin (AACT, >2-fold). Fold change generally in good agreement between the two anesthesia types, yet showed variability for CRP and SAA1 proteins (35.83 vs. 70.09 and 9.05 vs. 110.69 for PROP and SEVO, respectively), possibly due to low abundance of the proteins (close to noise signal level) in T0. In addition, there were four proteins specific to PROP and 7 proteins specific to SEVO which showed under a 2-fold change, with the exception of mannosidase alpha Class 1A Member 1 (MAN1A1, 2.12-fold change) in PROP and SAA2 in SEVO (48.67-fold change) (Figure 1 and Figure 2).

Next, proteomics data were validated by ELISA in both propofol- and sevoflurane-treated groups (Figure 3A,B). Acute phase response markers CRP and SAA1 were selected for validation. Both treatment groups showed an increase in line with the proteomic findings, with comparisons between T2 vs. T0 and T1 vs. T0 reaching statistical significance for both CRP and SAA1. In the case of CRP, ELISA detection showed an approximate 10-fold increase in both PROP and SEVO at T2 (Figure 3C), with SAA1 demonstrating an approximate 100-fold increase (Figure 3D).

### 2.3. Proteomics Profiles Independent of Anesthetic Regimen

As most of the changes in protein expression were found to be independent of the type of anesthetic, we combined both anesthesia groups (36 samples at three time points) for further analysis (Table 3). Seven proteins showed a change in abundance above two-fold (MTC *p* < 0.05). Table 3 also illustrates a clear indication of the changes in the levels of detected proteins over time. When comparing T1 vs. T0, 7 proteins showed statistical significance (MTC *p* < 0.05), increasing to 22 proteins in T2 vs. T1 and 28 when comparing T2 vs. T0.

## 3. Discussion

In this post hoc analysis of the VAPOR-1 trial, we analyzed the surgical stress response on the plasma proteomic level in healthy individuals undergoing a minimally invasive laparoscopic donor nephrectomy receiving either a propofol-based or a sevoflurane-based anesthesia. Currently, very little is known about the effect of any surgical intervention on the biological pathways involved in injury and repair. Our study provides a first step in this direction, since we were able to study the response in healthy individuals with a low comorbidity score whilst being exposed to the stress of a surgical procedure under anesthesia when donating one kidney for transplantation. The study of this unique ‘healthy’ cohort undergoing surgery may therefore serve as a ‘gold standard’ reference or control group for future research on how surgical and anesthetic intervention will affect outcomes in patients with acute or chronic diseases and significant comorbidities.

By using an unbiased proteome approach and analyzing individual sequential proteomic profiles, our results show that the highest changes in protein levels were found 24 h after surgery but appeared to be independent of the anesthetic agent used. Only some modest changes were detected between sevoflurane and propofol. There is conflicting evidence on which anesthetic agent is most beneficial for postoperative recovery [23,24]. Obviously, the optimal choice of an anesthetic for a particular surgical procedure or patient is the agent that will reduce the inflammatory response and thereby reduce the risk of postoperative morbidity after surgery [18,25].

A prominent category of proteins that we found upregulated included acute phase proteins (APPs) involved in the acute phase response (APR). APPs act as a first line of defense upon interruption of the homeostasis. APPs may function as pathogen recognition receptors (e.g., CRP, SAA, LPS, FGL-1), proteinase inhibitors (e.g., AACT, A1AT), or involve components of the complement, coagulation, and fibrinolytic system. APPs have diverse molecular functions that protect the host against pathogens and tissue injury [26].

CRP, a non-specific acute phase protein produced in the liver in response to IL-6 and other pro-inflammatory cytokines, plays a critical role in innate immunity by binding to ligands on apoptotic cells. This process activates the complement cascade and stimulates phagocytosis in immune cells [27]. CRP levels rise 6–8 h after surgical trauma, with a half-life of 19 h, and under normal conditions, peak 2–3 days after surgery [28,29]. In our study, the CRP level at baseline was low (3.4mg/L), but reached 53.9 mg/L one day after surgery, which demonstrates the minimally invasive nature of the procedure and is similar to increases in levels observed after laparoscopic hysterectomy or surgical repair of ankle fractures [30,31]. Significant changes in upregulation in CRP levels were found in both anesthetic groups, with less upregulation in the PROP group compared with the SEVO group (35- vs. 70-fold change, respectively). After ELISA validation, no differences in CRP levels between the two anesthetic groups were found, which suggests that the overall levels of injury and inflammation are similar.

SAA, especially the isoforms SAA1 and SAA2, are early highly sensitive acute phase proteins [32]. Plasma levels of SAA1/2 may increase up to 1000-fold or more 24–36 h after injury and remain elevated for 2–3 days [33,34]. SAA is considered an important protein in the acute phase response during inflammation. The exact physiological role, however, remains unclear. A generally accepted role of SAA in the APR is its role in cholesterol recycling and tissue remodeling through metalloproteinases [32]. After engulfment of cellular debris, macrophages are loaded with cholesterol. SAA-HDL binds to these macrophages and enhances the efflux of unesterified cholesterol. Unesterified cholesterol binds with SAA-free HDL, upon which HDL cholesterol can be redistributed and reused in inflammatory and repair mechanisms, while cholesterol-depleted macrophages continue phagocytosis [35]. SAA synthesis is induced by cytokines such as IL-1β, IL-6 and TNF-α. Linke et al. showed a critical role of SAA in survival; in a cecal ligation and puncture (CLP) model in normal mice, 75% of the wild type mice survived at day 5, while SAA-deficient or antibody-treated mice had a 90% mortality rate [36]. Of interest is the fact that, based on proteomic data, SAA1 levels in sevoflurane-treated individuals increased to a higher extent (110.69-fold change) compared to propofol-treated individuals (9.05-fold change), and that SAA2 only reached significant upregulation in sevoflurane-treated individuals and not in the propofol group. Based on ELISA validation, plasma SAA1 levels increased 677-fold one day after surgery, surpassing CRP levels, indicating SAA1 may serve as a more sensitive marker reflecting severity of tissue injury. Currently, it is unknown whether the amount of increase in SAA early in the APR indicates increased injury or enhanced repair capacity. However, persistently high levels of SAA after surgery have been associated with postoperative infections and adverse outcome [37,38].

LBP plays a dual role in the innate immune response to bacterial infection and inflammation. Baseline concentration (5–15 µg/L) can increase up to 30–50-fold during the acute phase response [39,40]. At low concentrations, LBP binds with lipopolysaccharide (LPS), a prominent component of Gram-negative bacteria, facilitating its recognition by membrane-bound CD14 (mCD14) on the surface of macrophages. Binding of the LPS-LBP complex to mCD14 will lead to the upregulation of transcription of pro-inflammatory cytokines like TNF-α and IL-1β [40]. For mCD14-negative cells, LBP transfers LPS to secreted CD14 (sCD14), which is recognized by membrane bound receptors, inducing transcription of pro-inflammatory cytokines like IL-6 and IL-8 [40]. In addition to its classic role, LBP can bind bacteria- and pathogen-associated molecular patterns (PAMPs). At high concentration, LBP neutralizes bacterial endotoxins and down-regulates the expression of TNF-α, preventing an overwhelming immune reaction [39,40]. LBP increases significantly upon severe infections and has been suggested as a biomarker for prediction of severe sepsis, particularly in the first 48 h [41]. The exact role of LBP in the perioperative phase, however, is unclear, with limited studies showing peak levels 72 h after appendectomy or colorectal surgery [42,43]. In cardiac surgery, no correlation was found between levels of LBP and IL-6 or CRP, with surprisingly no differences between on-pump and off-pump procedures, whereas higher levels were expected in on-pump patients due to lower perfusion pressure of the splanchnic circulation and higher likelihood of translocation of bacteria [43]. This suggests that LBP production is stimulated by another still unknown danger pattern which is identical in both on-pump and off-pump procedures. In the current study, LBP was significantly upregulated one day after surgery, with no differences between the PROP and SEVO group (both >4-fold change).

Both FGL-1 and A2GL were similarly upregulated under both anesthesia regimens. FGL-1, a member of the fibrinogen family, is produced by the liver upon stimulation by IL6 [44]. Its exact role in the APR to date is unknown. Since it is highly upregulated upon liver injury, FGL-1 was primarily linked to liver regeneration, until Liu and colleagues showed that enhanced expression of FGL-1 by IL-6 in the absence of liver injury suggests a role of FGL-1 in the APR [44]. Rijken et al. showed that FGL-1 is associated with fibrin matrix formation, indicating a potential function in regulating fibrin polymerization [45]. Additionally, FGL-1 was found upregulated in heart-transplanted rats with induced bacterial pulmonary infection [46]. To the best of our knowledge, this study is the first to show FGL-1 upregulation in healthy subjects undergoing a non-hepatic surgery.

Similarly to FGL-1, the exact biological function of A2GL currently is unclear. A2GL is expressed by hepatocytes, neutrophils, macrophages, and epithelial cells in response to cytokines like IL-6, IL-1β, and TNF-α. A2GL is involved in cell adhesion, differentiation of polymorphic nuclear cells, angiogenesis, and cell migration [47]. Oncogenic and tumor-suppressing properties have been reported, and A2GL is upregulated in many inflammatory diseases, such as rheumatoid arthritis [48]. In a murine epidermal injury model, Gao and colleagues showed that A2GL expression is high during wound repair [49]. In patients undergoing coronary artery bypass grafting (CABG), levels of A2GL are significantly upregulated 24 h after surgery [50]. The current study showed significant upregulation one day after surgery in both FGL-1 (>10-fold) and A2GL (>2-fold). The precise role of FGL-1 and A2GL in the APR, whether pro- or anti-inflammatory, or related to injury or repair, remains unclear. Dynamics of these proteins in patients with and without postoperative complications would therefore be an interesting area of study.

Another protein that was found significantly upregulated in an anesthesia-independent manner was alpha-1-antichymotrypsin (AACT), a serine protease inhibitor (serpin). Primarily, serpins neutralize overexpressed serine proteinase activity, thus controlling a variety of biological processes such as coagulation and inflammation [51]. AACT, the most abundant serpin in human plasma, inhibits a variety of serine proteases of which cathepsin G is thought to be its major target [50]. Cathepsin G is released by neutrophils at the site of inflammation or injury, where it activates pro-inflammatory cytokines, enhances coagulation and thrombosis, and kills and degrades pathogens [50]. AACT expression is modulated by, amongst others, IL-6 and IL-1, and forms a feedback loop controlling the APR. Banfi and colleagues showed that levels of AACT peak 24 h after CABG with an almost three-fold upregulation [50]. The upregulation of AACT, along with less abundant A1AT and metallopeptidase inhibitor 1 (TIMP1), an inhibitory molecule that regulates matrix metalloproteinases, in our living kidney donors one day after surgery suggests an increased anti-inflammatory response [52,53]. The observed downregulation of serine protease inhibitor-IPSP might be explained by enhanced consumption as a result of proteolysis after surgery.

Proteomic profiles displayed a number of moderate-level anesthesia-specific changes, with propofol eliciting a larger number of upregulated proteins than sevoflurane. SAA2 was only significantly upregulated in SEVO anesthesia. MAN1A1 was found to be upregulated specifically in PROP anesthesia. MAN1A1 is involved in glycosylation, which is crucial for stability and function of glycoproteins, including cytokines. MAN1A1 overexpression could enhance the expression of pro-inflammatory cytokines, potentially leading to dysregulation of the perioperative stress response [54].

Although discovering novel molecular insights into post-operative and anesthetic effects, our post hoc analysis has limitations. Since this was a pilot discovery study to identify proteomic changes, no power calculation was performed and the sample size was small. Therefore, the results of this study need to be validated in a larger prospective cohort. We observed differences between groups regarding depth of anesthesia and Cet of remifentanil during the procedure, with SEVO showing a higher average BIS and a lower Cet of remifentanil. Although we cannot exclude that this has interfered with our results, we consider these differences small and not clinically relevant. In addition, lactate levels before explanting the kidney in the SEVO group were higher than in the PROP group. Both groups, however, were within the normal physiological range. Since our living kidney donors were mainly discharged from the hospital on the third day after surgery, we were unable to study the proteomic changes over a longer period of time. In addition, only two donors developed postoperative complications. Therefore, caution is needed when making correlations between the levels of proteins and outcome.

In summary, we have used an unbiased proteomic approach to assess changes in molecular proteomic profiling due to surgical intervention in healthy individuals who are donating a kidney for transplantation. The types and quantities of proteins identified suggest a significant upregulation of the acute phase response, activation of the coagulation cascade, and initiation of tissue regeneration within 24 h after surgical intervention, which is independent of the anesthetic agent used. The magnitude and timing of the stress response is compatible with the moderate invasive character of the surgical procedure and consistent with the low incidence of postoperative complications and fast recovery observed in our study. Our proteomic results also suggest that there are delicate balances between pro- and anti-inflammatory acute phase responses, independent of the anesthetic agent used. It is important that after 24 h, proteins can be identified that suggest the presence of tissue regeneration and point at a subtle balance between injury and repair in the early postoperative period.

In conclusion, proteomic profiling identified upregulated proteins one day after laparoscopic donor nephrectomy. The proteins identified in this study may serve as a reference when developing a profile of potential markers to better monitor the immune response and effect of postoperative medical treatment, attempting to reduce surgery-related complications and improving short-and long-term outcomes.

## 4. Materials and Methods

### 4.1. Study Population

This study is a post hoc analysis of Volatile Anesthetic Protection of Renal transplants (VAPOR)-1 trial (NCT01248871). Stored plasma samples of donors participating in the VAPOR-1 trial were used. The VAPOR-1 trial is a prospective randomized controlled trial on the effects of two different anesthetic agents (propofol vs. sevoflurane) on renal outcome in living donor kidney transplantation [55]. The Institutional Review Board of the University Medical Centre of Groningen approved the study protocol of VAPOR-1 (METc 2009/334), which was conducted in adherence to the Declaration of Helsinki and registered with ClinicalTrials.gov: NCT01248871. Of the 57 donors, 19 patients received a sevoflurane-based anesthesia and 38 a propofol-based anesthesia. For this post hoc analysis we selected donors without any or with only minor comorbidities and included 17 donors of the sevoflurane group (SEVO) and 19 matched donors of the propofol group (PROP) with a complete set of plasma samples. The two groups were matched for age, gender, body mass index (BMI), concomitant comorbidities, medication use, and smoking status.

### 4.2. Surgery and Anesthesia

Kidney donation was performed via a hand-assisted laparoscopic procedure under general anesthesia. The choice of anesthetic agent (propofol or sevoflurane) was based on randomization. Intraoperative analgesia was managed with remifentanil with the use of target-controlled infusion. The depth of anesthesia, fluid administration, hemodynamic management, and administration of all medications were strictly protocolized. Postoperative pain management on day 1 encompassed paracetamol 1000 mg 4 times daily and intravenous piritramide with the use of patient-controlled analgesia [55].

### 4.3. Samples Acquisition

Blood samples in the VAPOR-1 study were taken at standardized time points [55]. For this project, we analyzed samples taken at three different time points (T0-T2): T0—baseline before induction of anesthesia; T1—end of surgery upon skin closure, and T2—24 h after surgery. Samples were centrifuged (1500× *g*, 20 min) and plasma was collected and stored at −80 °C until analysis.

### 4.4. Sample Preparation

All 108 samples involved in the study were first subjected to depletion of the top 12 most abundant proteins using T12 Depletion (Pierce™ Top 12 Abundant Protein Depletion Spin Columns (Thermo Fisher Scientific, Waltham, MA, USA). To this end, 10 µL of sample was incubated with gentle mixing for 60 min at room temperature and then centrifuged at 1000× *g* for 2 min. The filtrate containing the depleted sample was suspended in 10 mM phosphate-buffered saline (PBS), NaCl, 0.02% azide, pH 7.4. Protein concentration was assessed by Bicinchoninic acid (BCA) protein assay kit (Thermo Fisher Scientific, Waltham, MA, USA).

A measure of 20 µg of T12-depleted sample was then digested using the SMART method. Briefly, each sample was loaded into a SMART digestion tube (Thermo Fisher Scientific, Waltham, MA, USA) containing 150 µL of SMART digestion buffer. These were then incubated at 70 °C at 1400 revolutions per min (rpm) for 1 h on a heat shaker (Eppendorf, Hamburg, Germany) and then spun at 2500× *g* for 5 min, collecting the supernatant containing tryptic peptides. The samples were desalted using SOLAu SPE plates (Thermo Fisher Scientific, Waltham, MA, USA). Columns were equilibrated using 100% acetonitrile, then 0.1% trifluoroacetic acid (TFA). The samples were then acidified with 1% TFA and pulled through the column using a vacuum pump. They were then washed with 0.1% formic acid (FA) and eluted using 100 µL of 65% acetonitrile. Eluted samples were dried using a vacuum concentrator (SpeedVac, Thermo Fisher Scientific, Waltham, MA, USA) and resuspended in buffer A (98% MilliQ-H2O, 2% ACN, 0.1% FA) for mass spectrometry analysis.

### 4.5. Mass Spectrometry Analysis

For peptide analysis, nano-liquid chromatography tandem mass spectrometry (LC-MS/MS) was used, consisting of ultra-high performance liquid chromatography (uHPLC) coupled to a hybrid quadrupole-Orbitrap mass spectrometer (Fusion LUMOS, Thermo Fisher Scientific, Waltham, MA, USA), as described previously [56]. In brief, 1 µL of 0.5 µg/µL of peptide material was injected for analysis by LC-MS/MS. Peptides were separated by an Easy-Spray LC C18 column (75 µm × 50 cm, Thermo fisher Scientific, Waltham, MA. USA) at a flow rate of 250 nL/min. The mobile phases consisted of water with 0.1% FA, 5% DMSO (buffer A) and 95% acetonitrile with 5% DMSO, 0.1% FA (buffer B), respectively. A 60 min linear gradient from 3% buffer A to 40% buffer B was used. The peptides were ionized by electro spray ionization and the 20 most abundant ions per MS scan were fragmented by collision-induced dissociation (CID), as described previously [57].

### 4.6. Data Analysis and Statistics

MaxQuant software (v1.5.8.3) was used for processing raw MS data and for peptide and protein identification and quantitation. LC-MS/MS spectra were searched against the UniProt human database (version 2017, 20,205 entries) for peptide homology identification. At least one unique peptide was used for protein quantitation using match between runs. The false discovery rate (FDR) was set to 1% for protein and peptide identification. Label-free quantitation (LFQ) intensity data were used for further statistical analysis to compare across the different time points (T2 vs. T1 and T0) and groups (propofol vs. sevoflurane). Differentially expressed proteins in the analysis were defined as proteins presenting a statistical difference across the group (*p* < 0.05) (proteins detected in at least three individuals/group to allow Student’s *t*-test). Multiple testing correction (MTC) was applied for comparisons between time points in both anesthesia groups. UniProt and Panther (gene ontology) databases were utilized for the interpretation of the molecular function of identified targets.

### 4.7. ELISA

To validate proteomic findings, ELISA was used to detect CRP and serum amyloid A (SAA)1 (R&D Systems, Minneapolis, MN, USA, CRP DuoSet, SAA1 DuoSet). Plates were coated with capture antibody at room temperature overnight and blocked with Reagent Diluent for 1 h at room temperature (RT). A measure of 100 µL of diluted plasma sample was added for 2 h at RT, followed by antibody detection for 2 h, then Horseradish peroxidase (HRP) and substrate solution for 20 min each at RT. Stop solution was then added and the plate was analyzed using a TECAN plate reader at a wavelength of 540 nm.

## Figures and Tables

**Figure 1 ijms-26-05196-f001:**
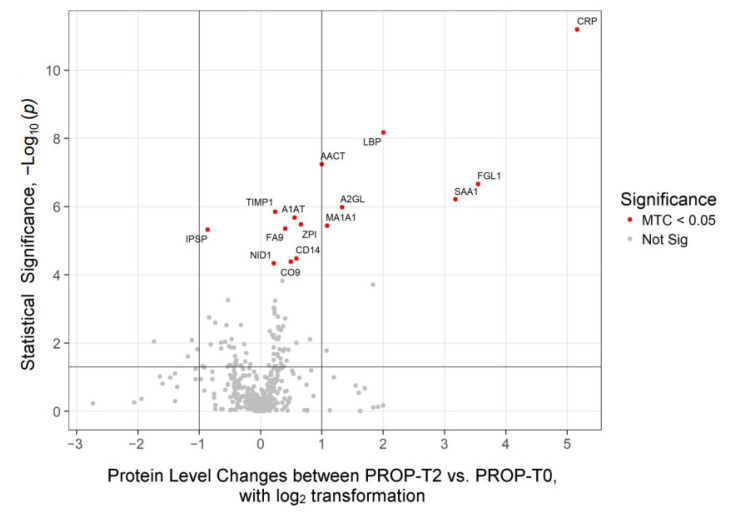
Propofol exposure affects the plasma proteome pre- versus post-operation. Volcano plot showing changes in protein abundance levels between PROP-T2 vs. PROP-T0. Abundance thresholds are set to 1 and −1 (log2), respectively. A multiple test correction (MTC) was applied to highlight plasma proteins with significantly altered levels (marked as red dots).

**Figure 2 ijms-26-05196-f002:**
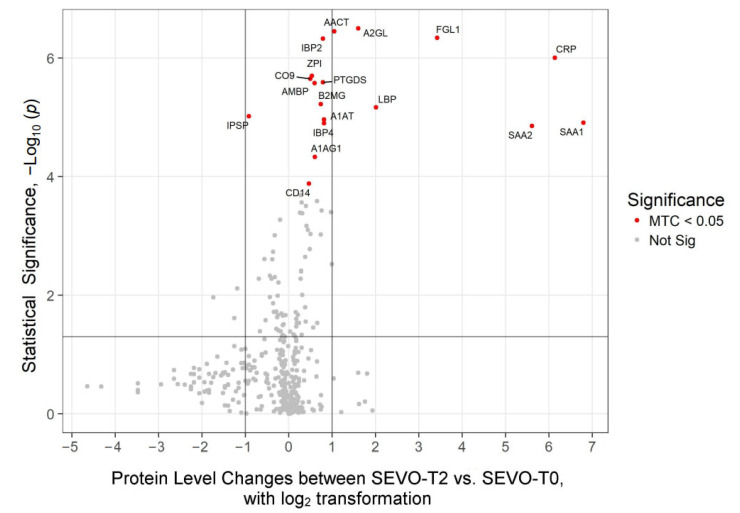
Sevoflurane exposure affects the plasma protein pre- versus post-operation. Volcano plot showing changes in protein abundance levels between SEVO-T2 vs. SEVO-T0. Abundance thresholds are set to 1 and −1 (log2), respectively. A multiple test correction (MTC) was applied to highlight plasma proteins with significantly altered levels (marked as red dots).

**Figure 3 ijms-26-05196-f003:**
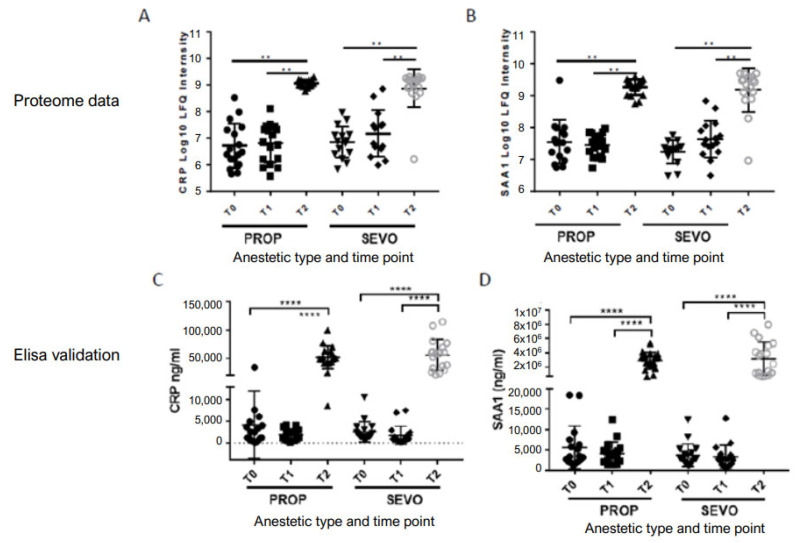
Global changes in plasma proteome levels induced by exposure to both anesthetics. Proteome levels of C-reactive proteins (CRP) (**A**) and serum amyloid-A1 (SAA1) (**B**) in both propofol and sevoflurane at three time points. ELISA validation of CRP (**C**) and SAA-1 (**D**) proteome levels in both propofol and sevoflurane at three time points. ** *p* < 0.01 (unpaired two-sided test); **** *p* < 0.0001 (unpaired two-sided *t* test).

**Table 1 ijms-26-05196-t001:** Patients’ demographics and peri-operative data in PROP and SEVO anesthesia groups. Data are given as *n* (%), mean (SD), or median (IQR).

	PROP *n* = 19	SEVO *n* = 17	*p*
Baseline characteristics			
Age y	49.6 (13.2)	33.8 (10.6)	0.297
Male *n* (%)	10 (53%)	9 (53%)	>0.999
BMI kg/m^2^	25.9 (3.6)	27.4 (3.5)	0.233
ASA I/II	15/4	11/6	0.463
mGFR mL/kg	125 (98–140)	107 (97–132)	0.364
Smoking *n* (%)	7 (37%)	5 (30%)	0.728
CCI	1 (0–2)	1 (0–1)	0.620
Cardiovascular comorbidity *n* (%)	1 (5%)	5 (29%)	0.080
MAP baseline mmHg	93 (90–96)	94 (85–104)	0.863
Perioperative data			
Duration procedure min	230 (28.4)	245 (42.5)	0.196
Amount of fluid mL/kg BW	61.5 (10.0)	60.2 (11.7)	0.730
BIS	39 (7)	45 (6)	0.012
Blood sample clip renal artery			
pH	7.41 (0.04)	7.39 (0.04)	0.146
PaO2 kPa	20.3 (4.9)	20.4 (4.7)	0.955
Hemoglobin mmol/L	7.2 (0.9)	7.3 (1.0)	0.725
Lactate mmol/L	1.3 (0.5)	1.7 (0.7)	0.044
Anesthetics/analgesics			
Propofol Cet (μg/mL)	3.3 (0.5)	-	-
Sevoflurane EtC	-	1.53 (0.14)	-
Remifentanil Cet (ng/mL)	3.2 (0.86)	2.6 (0.63)	0.020
Vasoactive medication			
Ephedrine *n* (%)	13 (68%)	17 (100%)	0.020
Dose mg	15 (10–20)	15 (10–30)	0.207
Phenylephrine *n* (%)	2 (11%)	1 (6%)	>0.999
Dose μg	150 (100–200)	100	*n* too small
Other medication			
Piritramide intraoperative mg	8 (7.0–9.0)	7.5 (7.0–8.5)	0.901
Piritramide postoperative mg	12 (10.0–16.0)	11 (8.5–19.0)	0.688
Ondansetron intraoperative *n* (%)	2 (11%)	8 (47%)	0.025
Ondansetron PACU *n* (%)	8 (42%)	3 (18%)	0.156
Dexamethasone *n* (%)	2 (11%)	3 (18%)	0.650
Droperidol *n* (%)	1 (5%)	1 (6%)	>0.999
Postoperative complications	2 (10%)	0 (0%)	0.487
LOH *d*	5 (5–7)	5 (5–7.5)	0.912

*n*—number in group; BMI—body mass index; ASA—classification American Society for Anesthesiology; mGFR—Glomerular Filtration Rate, measured by isotope 125I-iothalamate; CCI—Charlson comorbidity index; MAP—mean arterial pressure; BW—body weight BIS—bispectral index; Cet—effect side concentration; EtC—end tidal concentration; PACU—post-anesthetic care unit; LOH—length of hospital stay; *d*—days.

**Table 2 ijms-26-05196-t002:** Proteins displaying plasma level changes in patients stratified by the type of anesthesia (PROP and SEVO) between time points T2-vs-T0. Statistical differences across the groups (*p* < 0.05) (proteins detected in at least three individuals/group) were examined using Student’s *t*-test. Multiple testing correction (MTC) was applied for comparisons between time points in both anesthesia groups.

UniProt ID	Gene	Protein	PROP (T2-vs-T0)			SEVO (T2-vs-T0)		
			MTC Level	*p*-Value	Fold Change	MTC Level	*p*-Value	Fold Change
P02763	*A1AG1*	Alpha-1-acid glycoprotein 1	-	-	-	*p* < 0.05	4.60 × 10^−5^	1.52
P01009	*A1AT*	Alpha-1-antitrypsin	*p* < 0.01	2.10 × 10^−6^	1.47	*p* < 0.01	1.10 × 10^−5^	1.76
P02750	*A2GL*	Leucine-rich alpha-2-glycoprotein	*p* < 0.01	1.00 × 10^−6^	2.51	*p* < 0.01	3.20 × 10^−7^	3.04
P01011	*AACT*	Alpha-1-antichymotrypsin	*p* < 0.01	5.70 × 10^−8^	2	*p* < 0.01	3.50 × 10^−7^	2.07
P02760	*AMBP*	Protein AMBP	-	-	-	*p* < 0.01	2.60 × 10^−6^	1.51
P61769	*B2MG*	Beta-2-microglobulin	-	-	-	*p* < 0.01	6.00 × 10^−6^	1.67
P08571	*CD14*	Monocyte differentiation antigen CD14	*p* < 0.05	3.30 × 10^−5^	1.5	*p* < 0.05	1.30 × 10^−4^	1.38
P02748	*CO9*	Complement component C9	*p* < 0.05	4.10 × 10^−5^	1.41	*p* < 0.01	2.20 × 10^−6^	1.41
P02741	*CRP*	C-reactive protein	*p* < 0.01	6.40 × 10^−12^	35.83	*p* < 0.01	9.90 × 10^−7^	70.09
P00740	*FA9*	Coagulation factor IX	*p* < 0.01	4.40 × 10^−6^	1.32	-	-	-
Q08830	*FGL1*	Fibrinogen-like protein 1	*p* < 0.01	2.20 × 10^−7^	11.69	*p* < 0.01	4.50 × 10^−7^	10.69
P18065	*IBP2*	Insulin-like growth factor-binding protein 2	-	-	-	*p* < 0.01	4.70 × 10^−7^	1.73
P22692	*IBP4*	Insulin-like growth factor-binding protein 4	-	-	-	*p* < 0.01	1.30 × 10^−5^	1.76
P05154	*IPSP*	Plasma serine protease inhibitor	*p* < 0.01	4.80 × 10^−6^	−1.82	*p* < 0.01	9.50 × 10^−6^	−1.89
P18428	*LBP*	Lipopolysaccharide-binding protein	*p* < 0.01	6.60 × 10^−9^	4.01	*p* < 0.01	6.80 × 10^−6^	4.04
P33908	*MAN1A1*	Mannosyl-oligosaccharide 1,2-alpha-mannosidase IA	*p* < 0.01	3.60 × 10^−6^	2.12	-	-	-
P14543	*NID1*	Nidogen-1	*p* < 0.05	4.60 × 10^−5^	1.16	-	-	-
P41222	*PTGDS*	Prostaglandin-H2 D-isomerase	-	-	-	*p* < 0.01	2.60 × 10^−6^	1.73
P0DJI8	*SAA1*	Serum amyloid A-1 protein	*p* < 0.01	6.10 × 10^−7^	9.05	*p* < 0.01	1.20 × 10^−5^	110.69
P0DJI9	*SAA2*	Serum amyloid A-2 protein	-	-	-	*p* < 0.01	1.40 × 10^−5^	48.67
P01033	*TIMP1*	Metalloproteinase inhibitor 1	*p* < 0.01	1.40 × 10^−6^	1.18	-	-	-
Q9UK55	*ZPI*	Protein Z-dependent protease inhibitor	*p* < 0.01	3.30 × 10^−6^	1.58	*p* < 0.01	2.00 × 10^−6^	1.45

**Table 3 ijms-26-05196-t003:** Changes in plasma proteomic profile, independent of the anesthesia type (PROP or SEVO). Listed in the table are proteins displaying statistically significant (MTC *p*-value < 0.05) plasma level changes in at least one of the comparisons (T2 vs. T0, T2 vs. T1, or T1 vs. T0) in the combined patient cohort of this study (36 patients including 19 PROP and 17 SEVO anesthesia cases). Statistical differences across the groups (*p* < 0.05) (proteins detected in at least three individuals/group) were examined using Student’s *t*-test. Multiple testing correction (MTC) was applied for comparisons between time points in both anesthesia groups.

			ALL T2-T0				ALL T2-T1				ALL T1-T0			
UniPro ID	Gene	Protein	MTC	*p*-Value	Fold Change	Direction	MTC	*p*-Value	Fold Change	Direction	MTC	*p*-Value	Fold Change	**Direction**
P02741	*CRP*	C-reactive protein	*p* < 0.01	2.7 × 10^−16^	46.42	UP	*p* < 0.01	2.4 × 10^−14^	22.61	UP	-	-	-	-
P01011	*AACT*	Alpha-1-antichymotrypsin	*p* < 0.01	3.4 × 10^−14^	2.03	UP	*p* < 0.01	2.9 × 10^−13^	1.93	UP	-	-	-	-
P18428	*LBP*	Lipopolysaccharide-binding protein	*p* < 0.01	1.5 × 10^−13^	4.04	UP	*p* < 0.01	1.8 × 10^−9^	3.73	UP	-	-	-	-
Q08830	*FGL1*	Fibrinogen-like protein 1	*p* < 0.01	1.6 × 10^−13^	11.76	UP	*p* < 0.01	6.8 × 10^−11^	8.48	UP	-	-	-	-
P02750	*A2GL*	Leucine-rich alpha-2-glycoprotein	*p* < 0.01	7.2 × 10^−13^	2.74	UP	*p* < 0.01	7.9 × 10^−12^	2.63	UP	-	-	-	-
Q9UK55	*ZPI*	Protein Z-dependent protease inhibitor	*p* < 0.01	3.1 × 10^−11^	1.52	UP	*p* < 0.01	3.5 × 10^−8^	1.41	UP	-	4.3 × 10^−2^	1.08	UP
P0DJI8	*SAA1*	Serum amyloid A-1 protein	*p* < 0.01	5.7 × 10^−11^	17.69	UP	*p* < 0.01	2.2 × 10^−11^	32.42	UP	-	-	-	-
P05154	*IPSP*	Plasma serine protease inhibitor	*p* < 0.01	1.0 × 10−^10^	1.85	DOWN	*p* < 0.01	8.1 × 10^−9^	1.69	DOWN	-	-	-	-
P01009	*A1AT*	Alpha-1-antitrypsin	*p* < 0.01	3.2 × 10^−10^	1.61	UP	*p* < 0.01	1.2 × 10^−11^	2.09	UP	*p* < 0.05	8.8 × 10^−5^	1.3	DOWN
P02748	*CO9*	Complement component C9	*p* < 0.01	4.2 × 10^−10^	1.41	UP	*p* < 0.01	3.1 × 10^−7^	1.25	UP	-	9.3 × 10^−3^	1.13	UP
P00740	*FA9*	Coagulation factor XI	*p* < 0.01	6.7 × 10^−9^	1.32	UP	-	1.5 × 10^−2^	1.1	UP	*p* < 0.05	2.8 × 10^−5^	1.2	UP
P0DJI9	*SAA2*	Serum amyloid A-2 protein	*p* < 0.01	9.1 × 10^−9^	4.58	UP	*p* < 0.01	2.7 × 10^−9^	16.47	UP	-	-	-	-
P33908	*MAN1A1*	Mannosyl-oligosaccharide 1,2-alhpa-mannosidase IA	*p* < 0.01	1.0 × 10^−8^	1.93	UP	*p* < 0.05	6.2 × 10^−5^	1.63	UP	-	-	-	-
P08571	*CD14*	Monocyte differentiation antigen CD14	*p* < 0.01	1.1 × 10^−8^	1.44	UP	*p* < 0.01	1.2 × 10^−6^	1.38	UP	-	-	-	-
P18065	*IBP2*	Insulin-like growth factor-binding protein 2	*p* < 0.01	1.2 × 10^−8^	1.38	UP	*p* < 0.01	2.9 × 10^−7^	1.47	UP	-	2.6 × 10^−2^	1.06	DOWN
P15169	*CBPN*	Carboxy peptidase N catalytic chain	*p* < 0.01	1.9 × 10^−6^	1.25	UP	-	8.0 × 10^−4^	1.22	UP	-	-	-	-
P02760	*AMBP*	Protein AMBP	*p* < 0.01	2.9 × 10^−6^	1.4	UP	*p* < 0.05	2.6 × 10^−5^	1.22	UP	-	-	-	-
P01033	*TIMP1*	Metalloproteinase inhibitor 1	*p* < 0.01	3.2 × 10^−6^	1.23	UP	*p* < 0.01	1.4 × 10^−5^	1.24	UP	-	-	-	-
P36955	*PEDF*	Pigment epithelium-derived factor	*p* < 0.01	3.3 × 10^−6^	1.27	UP	*p* < 0.01	1.1 × 10^−6^	1.21	UP	-	-	-	-
P05019	*IGF1*	Insulin-like growth factor I	*p* < 0.01	8.5 × 10^−6^	1.25	UP	-	-	-	-	-	1.3 × 10^−2^	1.14	UP
P04004	*VTNC*	Vitronectin	*p* < 0.01	1.4 × 10^−5^	1.22	UP	-	-	-	-	-	2.0 × 10^−2^	1.13	UP
P49747	*COMP*	Cartilage oligomeric matrix protein	*p* < 0.01	1.5 × 10^−5^	1.47	DOWN	-	1.6 × 10^−2^	1.2	DOWN	-	7.1 × 10^−4^	1.2	DOWN
P00748	*FA12*	Coagulation factor XII	*p* < 0.05	2.6 × 10^−5^	1.32	DOWN	-	1.5 × 10^−3^	1.27	DOWN	-	-	-	-
P02787	*TRFE*	Serotransferrin	*p* < 0.05	3.2 × 10^−5^	1.54	DOWN	-	-	-	-	-	7.0 × 10^−3^	1.3	DOWN
P02763	*A1AG1*	Alpha-1-acid glycoprotein 1	*p* < 0.05	3.4 × 10^−5^	1.38	UP	*p* < 0.01	3.3 × 10^−10^	1.7	UP	-	2.6 × 10^−2^	1.23	DOWN
Q86UD1	*OAF*	Out at first protein homolog	*p* < 0.05	4.1 × 10^−5^	1.87	UP	*p* < 0.01	8.1 × 10^−6^	1.67	UP	-	-	-	-
P02649	*APOE*	Apolipoprotein E	*p* < 0.05	8.0 × 10^−5^	1.25	UP	-	-	-	-	*p* < 0.01	6.5 × 10^−6^	1.31	UP
P22792	*CPN2*	Carboxy peptidase N subunit 2	*p* < 0.05	8.3 × 10^−5^	1.16	UP	-	-	-	-	-	-	-	-
Q92820	*GGH*	Gamma-glutamyl hydrolase	-	1.5 × 10^−4^	1.15	UP	-	-	-	-	*p* < 0.05	9.0 × 10^−5^	1.15	UP
P14543	*NID1*	Nidogen-1	-	2.5 × 10^−4^	1.05	UP	*p* < 0.01	3.2 × 10^−10^	1.08	UP	-	-	-	-
P05160	*F13B*	Coagulation factor XIII B chain	-	1.9 × 10^−3^	1.39	DOWN	*p* < 0.05	7.9 × 10^−6^	1.59	DOWN	-	-	-	-
P22352	*GPX3*	Glutathione peroxidase 3	-	2.8 × 10^−3^	1.2	UP	-	-	-	-	*p* < 0.01	6.5 × 10^−6^	1.3	UP
Q06033	*ITIH3*	Inter-alpha-trypsin inhibitor heavy chain H3	-	1.2 × 10^−2^	1.17	UP	*p* < 0.01	1.9 × 10^−7^	1.19	UP	-	-	-	-
P22105	*TENX*	Tenascin-X	-	4.0 × 10^−2^	1.14	UP	-	-	-	-	*p* < 0.05	2.7 × 10^−5^	1.35	UP
O75636	*FCN3*	Ficolin-3	-	-	-	-	-	4.2 × 10^−3^	1.19	DOWN	*p* < 0.05	5.0 × 10^−5^	1.26	UP

## Data Availability

Clinical data generated during the study and used for the analysis are available in the data repository DataverseNL. This does not include sensitive patient data. Proteomics data generated during this study are available in the PRoteomics IDEntifications Database (PRIDE-EMBL-EBI) [58,59].

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
