# Peer review of "Effect of a Laparoscopic Donor Nephrectomy in Healthy Living Kidney Donors on the Acute Phase Response Using Either Propofol or Sevoflurane Anesthesia"

_ijms, 2025, doi:10.3390/ijms26115196_

Round 1

Reviewer 1 Report

Comments and Suggestions for Authors

Thanks for asking me to review this manuscript. 

Overall I thought that this was well written, comprehensive and relevant, however I do have some specific comments to make;

The title could be changed to make the manuscript more attractive to the reader - perhaps "The effects of healthy donor nephrectomy on the acute phase response using either propofol or sevoflurane".

The authors should make clear to the reader that this study is, in fact, a sub-analysis from a trial registered as VAPOR-1 (NCT01248871) and published in the BJA in 2017, 8 years ago. As such the current study was not powered to detect differences between the groups as this was not a primary outcome of the original study and the relatively small numbers of patients included make it difficult to draw specific conclusions.

For fellow researchers interested in the topic the data should be available as supplementary material, rather than "available on request".   

There are some minor grammatical and spelling errors which can be corrected during the typesetting.

Author Response

Comment 1
The title could be changed to make the manuscript more attractive to the reader - perhaps "The effects of healthy donor nephrectomy on the acute phase response using either propofol or sevoflurane".

Response 1
We agree with this comment. We changed the title in ‘Effect of a laparoscopic donor nephrectomy in healthy living kidney donors on the acute phase response using either propofol or sevoflurane anesthesia". (page 1, paragraph 1, lines 2-4)

Comment 2
The authors should make clear to the reader that this study is, in fact, a sub-analysis from a trial registered as VAPOR-1 (NCT01248871) and published in the BJA in 2017, 8 years ago. As such the current study was not powered to detect differences between the groups as this was not a primary outcome of the original study and the relatively small numbers of patients included make it difficult to draw specific conclusions.

Response 2
We clarified to the reader that this study is a sub-analysis from the VAPOR-1 trial by making some adjustments to the methods and discussion section:

Methods:
This study is a post-hoc analysis of Volatile Anaesthetic Protection Of Renal transplants (VAPOR)-1 trial (NCT01248871). Stored plasma samples of donors participating in the VAPOR-1 trial were used. The VAPOR-1 trial is a prospective randomized controlled trial on the effects of two different anesthetic agents (propofol vs sevoflurane) on renal outcome in living donor kidney transplantation.’  (page 14, paragraph 2, lines 362-366)

Discussion:
‘In this post-hoc analysis of the VAPOR-1 trial, we analyzed the surgical stress response on a plasma proteomic level in healthy individuals undergoing a minimally invasive laparoscopic donor nephrectomy receiving either a propofol-based or a sevoflurane-based anesthesia.’ (page 11, paragraph 1, lines 195-198)

Comment 3
For fellow researchers interested in the topic the data should be available as supplementary material, rather than "available on request".   

Response 3
We agree with this reviewer’s suggestion. According to this comment, we decided to use DataverseNL as a data repository to share our patient data openly. We will store a minimal dataset in DataverseNL which does not include personal/sensitive patient data.  The MS data will be stored in PRIDE.

  1. Proteomics data generated during this study will be deposited to the PRoteomics IDEntifications Database (PRIDE - EMBL-EBI) (REF)
  2. will be deposited to the DataverseNL platform (repository number will follow)

The data availability statement was changed:
Data Availability Statement: Clinical data generated during the study and used for the analysis is available in the data repository DataverseNL. This does not include sensitive patient data. Proteomics data generated during this study is available in the PRoteomics IDEntifications Database (PRIDE - EMBL-EBI) (REF) (page 16, paragraph 7, lines 467-470)

The following references were included: (page 21, lines 617-620)
58. Perez-Riverol Y, Csordas A, Bai J, et al. The PRIDE database and related tools and resources in 2019: improving support for  quantification data. Nucleic Acids Res. 2019;47(D1):D442-D450. doi:10.1093/nar/gky1106
59. Perez-Riverol Y, Bai J, Bandla C, et al. The PRIDE database resources in 2022: a hub for mass spectrometry-based proteomics evidences. Nucleic Acids Res. 2022;50(D1):D543-D552. doi:10.1093/nar/gkab1038

Comment 4
There are some minor grammatical and spelling errors which can be corrected during the typesetting.

Response 4
According to this comment, we checked the whole manuscript for grammatical or spelling errors and made adjustments in the manuscript. We uploaded a clean version of the revised manuscript and a version with tracked changes in supplementary files.

Reviewer 2 Report

Comments and Suggestions for Authors
  • The abstract seems not well completed. What kind of implications will happen if the proteins are upregulated. The research significance should be emphasized in the end.
  • There are too many keywords. I will recommend to reserve six of them.
  • In Table 2, does MTC level indicates the same meaning of P-Value? It seems repetitive.
  • In Table 2, what was the reason to analyze these selected genes and proteins? Did any research studied the correlations?
  • Mos of the references are too old. The authors need to cite more literature published in the past five years.

Author Response

Comment 1
The abstract seems not well completed. What kind of implications will happen if the proteins are upregulated. The research significance should be emphasized in the end.

Response 1
According to this comment, the abstract is rewritten:
‘Surgical trauma elicits a complex inflammatory stress response, contributing to postoperative morbidity and recovery variability. This response is influenced by patient-specific factors and surgical and anesthetic techniques. To isolate the impact of anesthesia on the acute phase response, we investigated plasma proteomic changes in a uniquely homogeneous cohort of healthy living kidney donors (n = 36; propofol = 19, sevoflurane = 17) undergoing laparoscopic donor nephrectomy. Proteomic profiling of plasma samples collected preoperatively and at 2 and 24 hours postoperatively revealed 633 quantifiable proteins, of which 22 showed significant perioperative expression changes. Eight proteins exhibited over two-fold increases, primarily related to the acute phase response (CRP, SAA1, SAA2, LBP), tissue repair (FGL1, A2GL), and anti-inflammatory regulation (AACT). These changes were largely independent of anesthetic type, though SAA2 and MAN1A1 showed anesthetic-specific expression. The upregulation of these proteins implicates activation of immune pathways involved in host defense, tissue remodeling, and inflammation resolution. Our findings provide a molecular reference for the surgical stress response in healthy individuals and highlight candidate biomarkers for predicting and managing postoperative outcomes. Understanding these pathways may support development of strategies to mitigate surgical stress and enhance recovery, particularly in vulnerable patient populations.’  (page1, paragraph 1, lines 20-38)

Comment 2
There are too many keywords. I will recommend to reserve six of them.

Response 2
According to this comment, we changed the keywords into:
Keywords: surgical stress response, propofol, sevoflurane, anesthesia, living kidney donors, laparoscopic surgery (page 1, paragraph 1, lines 39-40)

Comment 3
In Table 2, does MTC level indicates the same meaning of P-Value? It seems repetitive.

Response 3
We acknowledge the criticisms raised by the reviewer. This applies to Table 2 as well as Table 3. Statistical differences across the groups (P<0.05) (proteins detected in at least three individuals/group) were examined using a Student T-test). In addition, to enforce a more rigorous threshold, multiple testing correction (MTC) was applied for comparisons between time points in both anesthesia groups as described in the Materials and Methods section. To improve clarity, we have now added a short explanation in Tables 2 and 3 in the revised manuscript.

Methods: ‘Differentially expressed proteins in the analysis were defined as proteins presenting a statistical difference across the group (P<0.05) (proteins detected in at least three individuals/group to allow a Student T-test). Multiple testing correction (MTC) was applied for comparisons between time points in both anesthesia groups.’ (Page 15-16, paragraph 4, lines 435-438.)

Legend Table 2 and 3: ‘Statistical differences across the groups (P<0.05) (proteins detected in at least three individuals/group) were examined using a Student T-test). Multiple testing correction (MTC) was applied for comparisons between time points in both anesthesia groups.’  (Page 4-5, lines 150-153, and Page 8, lines 188-191)

Comment 4
In Table 2, what was the reason to analyze these selected genes and proteins? Did any research studied the correlations?

Response 4
As described in the method section, we used an unbiased approach which means the proteins are not selected upfront. The sample preparation, mass spectrometry analysis and the data analysis of the proteomic approach are described in the method section (page 15-16, paragraph 2-4, lines 390-438)

Comment 5
Most of the references are too old. The authors need to cite more literature published in the past five years.

Response 5
We replaced the references 4,7,9,13,16,29,31,33,42,46,51and 54 by the following references (page 17-20):

4. Ivascu, R., Torsin, L. I., Hostiuc, L., Nitipir, C., Corneci, D., & Dutu, M. (2024). The Surgical Stress Response and Anesthesia: A Narrative Review. Journal of clinical medicine13(10), 3017. https://doi.org/10.3390/jcm13103017

7. Margraf, A., Ludwig, N., Zarbock, A., & Rossaint, J. (2020). Systemic Inflammatory Response Syndrome After Surgery: Mechanisms and Protection. Anesthesia and analgesia131(6), 1693–1707. https://doi.org/10.1213/ANE.0000000000005175

9. Viikinkoski, E., Aittokallio, J., Lehto, J., Ollila, H., Relander, A., Vasankari, T., Jalkanen, J., Gunn, J., Jalkanen, S., Airaksinen, J., Hollmén, M., & Kiviniemi, T. O. (2024). Prolonged Systemic Inflammatory Response Syndrome After Cardiac Surgery. Journal of cardiothoracic and vascular anesthesia38(3), 709–716. https://doi.org/10.1053/j.jvca.2023.12.017

13. Simegn, G. D., Bayable, S. D., & Fetene, M. B. (2021). Prevention and management of perioperative hypothermia in adult elective surgical patients: A systematic review. Annals of medicine and surgery (2012)72, 103059. https://doi.org/10.1016/j.amsu.2021.103059

16. Cruz, F. F., Rocco, P. R. M., & Pelosi, P. (2021). Immunomodulators in anesthesia. Current opinion in anaesthesiology34(3), 357–363. https://doi.org/10.1097/ACO.0000000000000989

29. Mantovani, A., & Garlanda, C. (2023). Humoral Innate Immunity and Acute-Phase Proteins. The New England journal of medicine388(5), 439–452. https://doi.org/10.1056/NEJMra2206346

31. Sereda, A. P., Rukina, A. N., Trusova, Y. V., Dzhavadov, A. A., Cherny, A. A., Bozhkova, S. A., Shubnyakov, I. I., & Tikhilov, R. M. (2023). Dynamics of C-reactive protein level after orthopedic surgeries. Journal of orthopaedics47, 1–7. https://doi.org/10.1016/j.jor.2023.11.014

33. Lundin, E.S., Wodlin, N.B., Nilsson, L. et al.Markers of tissue damage and inflammation after robotic and abdominal hysterectomy in early endometrial cancer: a randomised controlled trial. Sci Rep10, 7226 (2020). https://doi-org.proxy-ub.rug.nl/10.1038/s41598-020-64016-1

42. Meng, L., Song, Z., Liu, A., Dahmen, U., Yang, X., & Fang, H. (2021). Effects of Lipopolysaccharide-Binding Protein (LBP) Single Nucleotide Polymorphism (SNP) in Infections, Inflammatory Diseases, Metabolic Disorders and Cancers. Frontiers in immunology12, 681810. https://doi.org/10.3389/fimmu.2021.681810

46. Turgunov, Y., Ogizbayeva, A., Shakeyev, K., Mugazov, M., Akhmaltdinova, L., Nuraly, S., & Rudolf, V. (2024). The dynamics of the lipopolysaccharide-binding protein (LBP) level in assessing the risk of adverse outcomes in operated colorectal cancer patients. Asian journal of surgery47(8), 3435–3441. https://doi.org/10.1016/j.asjsur.2023.08.101

51. Fujimoto, M., Hosono, Y., Serada, S., Suzuki, Y., Ohkawara, T., Murata, O., Quick, A., Suzuki, K., Kaneko, Y., Takeuchi, T., & Naka, T. (2024). Leucine-rich α2-glycoprotein as a useful biomarker for evaluating disease activity in rheumatoid arthritis. Modern rheumatology34(5), 1072–1075. https://doi.org/10.1093/mr/road112

54. Kellici, T. F., Pilka, E. S., & Bodkin, M. J. (2021). Small-molecule modulators of serine protease inhibitor proteins (serpins). Drug discovery today26(2), 442–454. https://doi.org/10.1016/j.drudis.2020.11.012